# Investigating the Effects of Alltech Crop Science (ACS) Products on Plant Defence against Root-Knot Nematode Infestation

**DOI:** 10.3390/microorganisms11071700

**Published:** 2023-06-29

**Authors:** Anusha Pulavarty, Ankit Singh, Kira Young, Karina Horgan, Thomais Kakouli-Duarte

**Affiliations:** 1Molecular Ecology and Nematode Research Group, Department of Applied Science, enviroCORE, Kilkenny Road Campus, South East Technological University (SETU), R93 V960 Carlow, Ireland; anusha.pulavarty@itcarlow.ie (A.P.); ankit.singh@itcarlow.ie (A.S.); kiradyoung@gmail.com (K.Y.); 2Alltech Bioscience Centre, A86 X006 Dunboyne, Ireland; khorgan@alltech.com

**Keywords:** ACS5075, ACS3048, *Meloidogyne javanica*, RT-PCR, 2D-gel electrophoresis

## Abstract

Two formulations of Alltech Crop Science products (ACS), a proprietary blend of fermentation products and plant extracts with micronutrients (ACS5075), and a microbial based product (ACS3048), were tested to understand (1) their impact on the tomato plant immune response and (2) whether they are priming a resistance response in plants against root knot nematodes (RKN). Research findings reported previously indicate that tomato plants pre-treated with ACS5075 and ACS3048 were found less sensitive to *Meloidogyne javanica* infection. In the current study, the expression of six defence-related genes (*PR-1*, *PR-3*, *PR-5T*, *ACO*, *CAT* and *JERF 3*), relative to a housekeeping gene, were monitored via RT-PCR. Results suggest that the treatment with ACS5075 enhanced *ACO* and *PR-1* gene expression levels, both post- treatment and post-infection with *M. javanica*. Reduced *M. javanica* infestation that was reported in the previous study could be attributed to the increased expression of these genes in the ACS5075-treated plants. Tomato plants treated with ACS3048, but without RKN infection, also demonstrated higher levels of *ACO* and *PR-1* gene expression. Subsequently, 2D-gel electrophoresis was performed to study the differential protein expression in leaf tissues of treated tomato plants in an effort to elucidate a possible mechanism of action for these products. Protein spot 1 was identified as ‘disease resistance protein RPP13-like’, protein spot 2 as ‘phosphatidylinositol 4-phosphate 5-kinase 2’, spot 3 as ‘protein SABRE like’ and protein spot 4 as ‘uncharacterized protein’. Overall research findings indicate that the ACS products could be used as plant immunity-boosting agents, as they play a significant role in the expression of certain genes and proteins associated with plant defence.

## 1. Introduction

Plant parasitic nematode (PPN) infections cause serious crop losses worldwide and therefore are a threat to world agriculture and economy. They are widely spread with a broad host range causing yield losses of about 30% in susceptible crop varieties (tomatoes, eggplants and melons; [1]). Plant parasitic nematodes are known to manifest different diseases, such as root-knots in tomatoes, cysts in potatoes, white tip disease in rice, stunted growth in wheat and many more [2]. Root-knot nematodes (RKN) are PPN that are most prevalent and highly damaging [3]. The four predominant species are *Meloidogyne arenaria*, *M. incognita*, *M. javanica* and *M. hapla* [2]. De Waele and Elsen [4] have reported that the difficulty in controlling the damage caused by *Meloidogyne* species is due to their broad host range and high reproductive rate.

The application of synthetic chemical nematicides has been known to have various environmental and cost implications for growers [5]. They also have long term impacts on human health, ground waters and animals [1], and for this reason, many synthetic chemical nematicides have been banned [6]. Biofumigation has been practiced as one of the successful approaches in the past few years to manage PPNs [7]; however, one major limitation with this approach is that the efficacy of the fumigant deteriorates over time due to various environmental factors, such as temperature, humidity and soil characteristics [7]. Some research groups have reported the application of extracts from cruciferous plants [8], such as rape, mustard, canola, cabbage and broccoli [8,9,10], for PPN management, but these treatments have been shown to cause root decay and eventually lead to rotting of the plant biomass, resulting in unpleasant odours in the fields [8]. Other successful approaches include the application of organic amendments, bionematicides and fermentation products to control PPN infestation. However, before applying any such products a thorough investigation is essential to study their impact on other non-targeted organisms and overall soil health. It is also essential to study the mode of action of various products before field application. Therefore, there is still scope for research and development to investigate sustainable approaches to manage PPN infestations.

Priming is a mechanism or approach through which a plant’s defence mode is activated to prepare the plant to respond better to an abiotic or biotic factor and thus to improve crop productivity [11]. The process of inducing plant defence to confer resistance against broad range of pathogens is referred to as systemic acquired resistance (SAR). The immune system in plants is generally associated with or regulated by low molecular weight phytohormones, such as salicylic acid (SA), jasmonic acid (JA) and ethylene (ET). Systemic acquired resistance is regulated by SA, in collaboration with pathogenesis-related (PR) genes to provide long-term resistance in plants against biotrophic pathogens. Jasmonic acid and ET are among other basic plant hormones that activate rhizobacteria-induced systemic resistance (ISR) in plants [12].

The treatment of a plant with an elicitor or compounds, such as hormones, fermentation mixtures, soil health products or nematicides, help trigger its priming phase and therefore prepares the plant to encounter an infection. Chester [13] initially reported the induction of plant defence through externally applied elicitor molecules. Indeed, this hypothesis was later proven correct through the application of SA to tobacco plants, which enhanced the expression of PR-genes and provided plant resistance against tobacco mosaic virus (TMV; [14]). The application of *Bacillus firmus* (I-1582 isolates) to tomato plants before transplantation leads to plant systemic resistance and reduced development of *Meloidogyne* eggs [15]. There are several PR-genes that become overexpressed in plants when treated with certain elicitor compounds [12]. This overexpression further helps in the activation of plant defence and thus mediates resistance against plant pathogens.

Previous research findings outlined the impact that ACS products had on reducing RKN infestation by inhibiting egg mass formation on tomato roots and also on improvements in tomato plant growth, by enhancing biomass production of the treated plants [16,17]. The objective of this study was to establish what impact the ACS products have on the plant immune response and whether they are priming a resistance response in plants against RKN. The expression of six defence-related genes, relative to the actin housekeeping gene (Table 1), was monitored via RT-PCR using samples of both leaf and root tissue of treated plants with and without nematode infection. Out of the six genes studied, three were PR-genes (*PR-1*, *PR-3*, *PR-5T*), while the other three genes were *ACO*, *CAT* and *JERF 3*. Protein expression in the leaf tissues of tomato plants treated with ACS products was also examined in an effort to elucidate a possible mechanism of action for the products.

## 2. Materials and Methods

### 2.1. Sourcing, Culturing and Inoculation of Nematodes to Tomato Plants

Dr. E. A. Tzortzakakis, Hellenic Agricultural Organization—DEMETER, Crete, Greece—kindly provided tomato (*Solanum lycopersicum* cv. Moneymaker) roots that were infected with *M. javanica*. These roots were carefully used for further culturing of the RKN species within a South East Technological University (SETU) plant growth room. Roots of 3–4-week-old tomato seedlings were infected with 5–7 egg masses/plant around the roots. In the plant growth room, all the plants were maintained at a constant temperature of 26 ± 2 °C, 70 ± 10% relative humidity (RH), and a 14 h day/10 h night cycle. To prevent white fly, aphid, spider mite and other insect infestations, organic insecticide (Neudorff^®^ Organic Bug and Larvae killer) was applied (1%) once every three weeks on the leaves and stems. After a 8–12-week interval, tomato roots were examined for nematode galling damage. Using a sterile scalpel and forceps, egg masses were removed from the roots and kept at 9 °C to prevent juvenile hatching. All RKN work, including all growing, experimentation and waste disposal, was performed with permission from the Plant Health Division of the Irish Department of Agriculture, Food, and the Marine (DAFM) and under rigorous quarantine and confinement regulations.

### 2.2. Treatment of Tomato Plants with ACS Products and Nemguard^TM^

Tomato (cv. Moneymaker) seeds were germinated under environmentally controlled glasshouse conditions at 32 ± 2 °C and a natural day/night cycle. The seeds were first sown on sterilized garden soil with the following characteristics: pH 7.6 ± 0.02, electrical conductivity, 540 µs/cm; clay (%), 8 ± 0.02; silt (%), 19 ± 0.5; sand (%), 73 ± 3.9; Na, 12.58 (mg Kg^−1^); P, 2.8 (mg Kg^−1^); K, 30.59 (mg Kg^−1^). The seedlings were grown for a period of 3–4 weeks until they attained a height of approximately 6–10 cm before being transplanted into individual cups (4.5 cm × 12 cm) for all further experiments. The cups contained approximately 150–200 g of sterilized soil and compost mixture in a ratio of 1:1.

Two sets of experiments were performed; in the first set, 4-week-old tomato plants were treated individually with 3% (*v*/*v*) ACS5075 or 3 g ACS3048 in separate cups only once and were harvested after 3, 5, 9 and 15 days. For treatment with ACS5075, 50 mL of a 3% solution was prepared using distilled water, and this solution was added directly into the pot containing soil with one tomato plant in it. For treatment with ACS3048, 3 g of the powdered product was added as layers in between the soil in each cup, and then, one tomato plant was planted in each cup. Tomato plants without any treatment were considered the untreated control (UC) for this set of experiments. The ACS concentrations were selected based on previous results from entomopathogenic nematode work [16] and an RKN study [17]. Six replications (individual plants) were established per treatment and per time interval. All plants at 3 days post-treatment (3 dpt) were individually uprooted to proceed with RNA extraction and then RT-PCR. Subsequently, plants were uprooted after each dpt to proceed with RNA extraction and then RT-PCR; the same was performed for the control plants as well.

Leaf and root tissue, separately, was ground to a fine powder using liquid nitrogen with a sterile porcelain pestle and mortar, and 100 mg per sample was immediately used for RNA extraction or kept at −80 °C until it was used. Twelve tissue samples (two from each plant; leaf and root) were collected for each time point and treatment. 

In the second set of experiments, tomato plants were also grown for a period of 4 weeks and after attaining an approximate height of 10 cm, plants were randomly selected and individually received the following treatments: six tomato plants grown without any treatment or nematode inoculation were considered the untreated + uninoculated control plants (UC). Six more tomato plants were infected with approximately 500 freshly hatched *M. javanica* juveniles (J2)/plant. Infection was carried out by gently releasing the nematodes into three holes made in the soil around the plant roots using a micropipette. These six plants were only infected but not treated with any products and therefore were considered inoculated untreated plants (IU). Six more tomato plants were initially treated with 3% ACS5075 and 3 g ACS3048 one week before infecting them with approximately 500 J2/plant. The infection in these plants was conducted as outlined above. These six plants were considered treated + inoculated test plants (T). Plants that were infected with RKN J2 were gently removed from the pots after 3, 7 and 15 dpi (days post-inoculation) to check the development of egg masses on the roots. The development of egg masses was observed only after 15 dpi, and therefore, all the plants (IU, UC and T) were harvested after 15 dpi, when leaf and root samples were taken for RNA isolation. All treatments were conducted with six replications, and a total of 12 tissue samples (two from each plant; leaf and root) were collected per treatment. 

### 2.3. RNA Extraction and Quantitative Real-Time Reverse PCR

Leaf and root tissues, collected post-treatment and post-inoculation, were either used immediately or stored at −80 °C until use. RNA was extracted from the leaf and root tissues (approx. 100 mg/sample) for all the collected samples using the RNA-easy plant mini kit (Qiagen, Manchester, UK). The quality of the extracted RNA was verified by performing electrophoresis on 1.0% agarose gels, under denaturing conditions, with gels containing 2.2 M formaldehyde, and visualized using ethidium bromide stain. The extracted RNA samples were stored at −80 °C.

The expression of six defence-related genes relative to the actin housekeeping gene (Table 1) was monitored via RT-PCR (Roche LightCycler^®^ 96 System, Switzerland) from samples of both leaf and root tissue of the plants. Expression of the genes was studied after 3, 5, 9 and 15 dpt and 15 dpi. PCR mixtures (20 μL final volume) contained RNase-free water, 1 μL (0.4 μM) each of forward and reverse primers, 5 μL (200–250 ng/μL) of RNA template (approximately 1000 ng/reaction) and 4 μL (5 × concentration) of EvoScript RNA SYBR^®^ Green I Master Mix (Roche Life Science, Switzerland). PCR cycling conditions were as follows: 60 °C for 15 min (reverse transcription); 95 °C for 10 min (pre-incubation and initial denaturation); 40–45 cycles (amplification) of 95 °C for 10 s, 58 °C for 30 s, 95 °C for 60 s, 25 °C for 60 s, 95 °C continuous (melting curve) and the final step at 40 °C for 30 s (cooling) following the manufacturer’s instructions. 

*Actin* was used as the housekeeping gene, as its expression in tomato tissues does not vary after nematode infestation [12]. Primers sets are described in Table 1. The relative fold-changes in gene expression were calculated using the 2^−ΔΔCT^ method [19]. The cycle threshold (C_T_ values) that were generated by the Roche machine were recorded individually. ΔC_T_ values were calculated by deducting the individual C_T_ values from that of the C_T_ values obtained for the respective *Actin* gene for each sample [ΔC_T_(a target sample) − ΔCT(*Actin*)]. Using a reference gene (*Actin*) as a standard, the ultimate outcome of this method is represented as the mean relative fold-change in target gene expression in a target sample compared to that in the untreated control, which acted as the calibrating sample. 

### 2.4. Treatment of Tomato Plants with ACS Products, Protein Extraction, SDS-PAGE and 2D-Gel Electrophoresis

Tomato plants were grown in greenhouse conditions as described previously. Four-week-old tomato plants were treated with ACS5075, ACS3048 and Nemguard^TM^ at 3% (*v*/*v*), 3 g and 13.3 mg/L, respectively, for 9 days. Nemguard^TM^ is a commercial organic nematicide containing 45% garlic extract. After 9 days of treatment, shoot tissues were harvested, ground to a fine powder using liquid nitrogen in a sterile porcelain pestle and mortar and were stored at −80 °C, if not immediately used, for protein extraction. Total protein was extracted from 0.5 g of tissues/sample using the Sigma Total protein extraction kit^®^ (USA), following the manufacturer’s instructions. Protein concentrations were estimated using a NanoDrop^TM^. All the treatments were conducted in triplicate.

Initially, approximately 4–5 mg of protein was loaded in each well in a 4–20% Mini-PROTEAN^®^ TGX™ Precast Protein Gel (Biorad, USA) to analyse and separate proteins present in the samples based on molecular weight using SDS-PAGE. Subsequently, extracted protein samples were also analysed via separation through two dimensions, isoelectric point (pI) and molecular weight (MW), by performing 2D gel electrophoresis using the Invitrogen ZOOM^®^ IPGRunner™ system (Carlsbad, CA, USA), following the manufacturer’s instructions [20]. Approximately 400 μg (155 μL of the total volume diluted with rehydration buffer) of total protein per sample was loaded onto each ZOOM^®^ strip (pH 3–12) for protein separation based on the isoelectric point. Upon rehydration overnight, the second-dimension separation based on molecular weight was performed using an Invitrogen™ NuPAGE™ 4–12% Bis-Tris ZOOM™ protein gel, 1.0 mm, IPG-well. Once the samples were run, the gels were stained using the Coomassie staining procedure [21]. After 2 h of staining, gels were destained with methanol and glacial acetic acid. Subsequently, protein differences in the gels were observed using the white background of a white/UV transilluminator [21].

### 2.5. Protein Spot Excision from the 2D-Gels and N-Terminal Protein Sequencing

Upon completion of the staining and de-staining procedure of all the individual gels, the gels were carefully placed onto the gel documentation system to observe the protein spots using white light transmission. The unique protein spots were identified by comparing the gels containing untreated control protein samples with the gels containing protein samples from the plants treated with ACS5075, ACS3048 and Nemguard^TM^. The unique protein spots were carefully excised from the gels using a sterile scalpel and were suspended into 1 mL of storage buffer (5% glacial acetic acid) in 1.5 mL centrifuge tubes. The protein spots were labelled based on their molecular weight. These protein spots were collected from multiple runs and were sent for identification to Alta Biosciences in the United Kingdom. Automated N-terminal sequence analysis was carried out on each of the samples. 

The amino acid sequences were blasted against the available sequences in the NCBI database using Protein BLAST (https://blast.ncbi.nlm.nih.gov/Blast.cgi?CMD=Web&PAGE_TYPE=BlastHome) (Accessed date: 20 April 2022). The sequences that showed query coverage of 100%, with a lower E-value and being in *S. lycopersicum*, were considered the expected protein from each spot.

### 2.6. Statistical Analysis

In the first set of experiments, results recorded from RT-PCR using RNA samples from tomato plants (n = 6) that were not infected with RKN were presented as means ± the standard error (SE). These values were calculated using 2^−ΔΔCt^ and indicate gene expression changes in tissue from ACS-treated plants relative to those from untreated control (UC) plants and the actin housekeeping gene. Each value is derived from a single RNA extraction. The y-axis in Figure 1 and Figure 2 indicates the mean fold-change compared to UC. It is a ratio. If the ratio was equal to 1, it means that the expression was not different from that of UC. If the ratio was more or less than 1, it indicates that the expression was different from that of UC. These values of each group (n = 6) were subjected to a non-parametric Kolmogorov–Smirnov test using IBM-SPSS, version 23 (*p* ≤ 0.05). An asterisk (*) in Figure 1 and Figure 2 indicates that the means are significantly different from the untreated control (*p* ≤ 0.05). In the second set of experiments, the results of differential gene expression after inoculating the tomato plants with *M. javanica* were subjected to analysis of variance (ANOVA) (*p* ≤ 0.05), using IBM-SPSS, version 23. The various bars in Figure 3 represent the expression of each defence gene (*PR-1*, *PR-3*, *PR-5*, *ACO*, *CAT* and *JERF 3*) in the root and leaf tissues of the tomato plants in untreated + uninoculated control (UC), inoculated untreated (IU) and treated + inoculated (T) groups relative to the housekeeping gene actin. 

## 3. Results

### 3.1. Gene Expression Analysis

The *PR-1* gene expression increased in leaf tissues of the tomato plants treated with ACS5075. The relative fold-change of *PR-1* expression compared to that of the untreated control plants at 9 dpt and 15 dpt was 4-fold higher when compared to 3 dpt (Figure 1a). However, the relative expression of *PR-1* in the root tissues was very low and below 1 indicating that it was downregulated compared to UC, but it increased in both the leaf and root tissues upon ACS3048 treatment (Figure 2a). With ACS3048 treatment, in the leaf tissues, at 9 and 15 dpt, *PR-1* gene expression was 3-fold higher when compared to 3 dpt. In root tissues, however, the *PR-1* gene expression was noted to be highest at 15 dpt when compared to all the other time points. At 15 dpt, the expression was 5-fold higher when compared to that at 3 dpt (Figure 2a).

**Figure 1 microorganisms-11-01700-f001:**
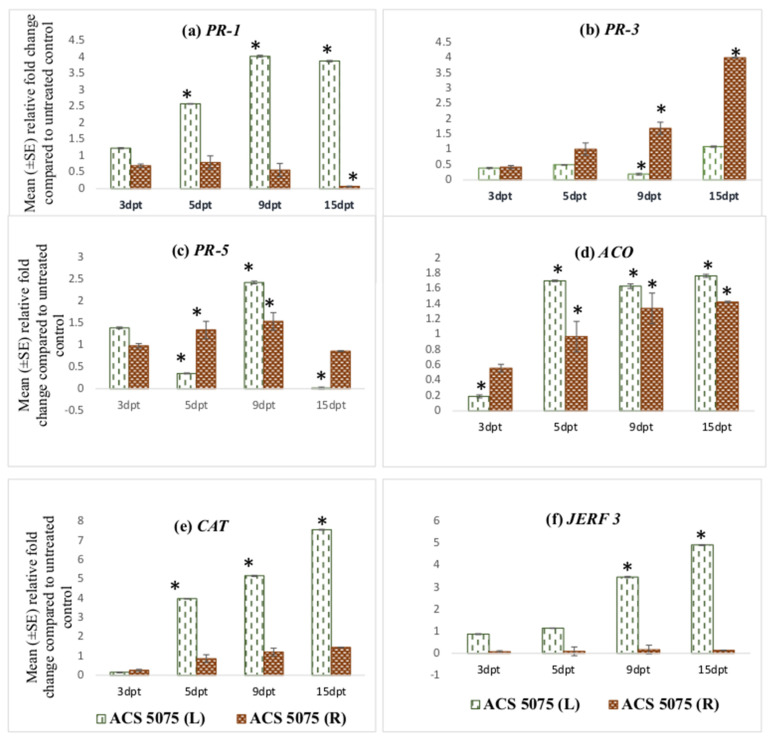
Effect of ACS5075 treatment on expression levels of (**a**) *PR-1*, (**b**) *PR-3*, (**c**) *PR-5*, (**d**) *ACO*, (**e**) *CAT* and (**f**) *JERF 3* genes in the leaf and root tissues of treated tomato plants compared to those in the untreated control (n = 6). (*) Symbol indicates a mean relative fold-change that is significantly different from that of the UC. Error bars indicate standard error reflecting variability within a sample (*p*-value ≤ 0.05). L refers to leaf tissues, and R refers to the root tissues, dpt refers to days post treatment.

**Figure 2 microorganisms-11-01700-f002:**
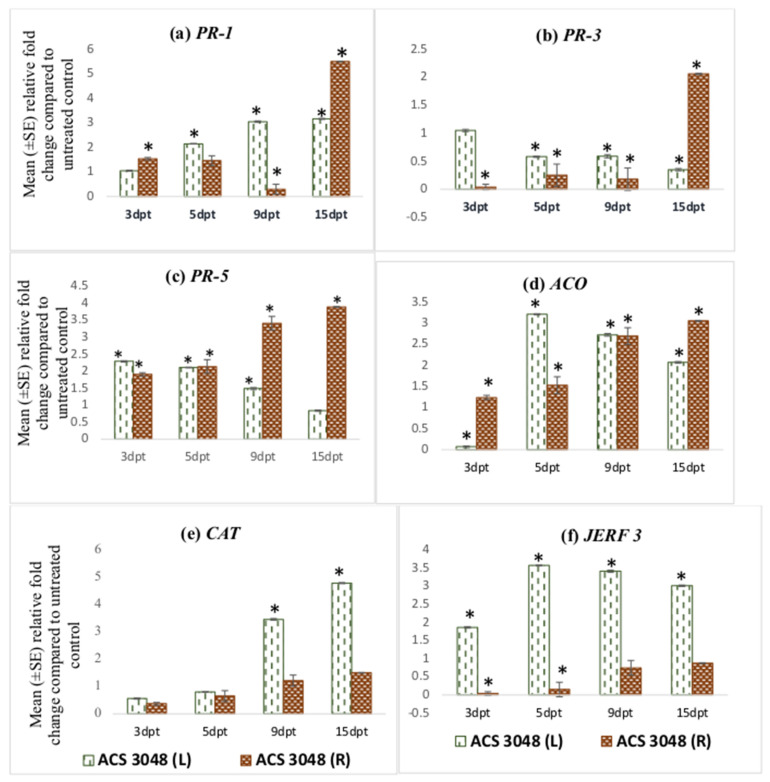
Effect of ACS3048 treatment on expression levels of (**a**) *PR-1*, (**b**) *PR-3*, (**c**) *PR-5*, (**d**) *ACO*, (**e**) *CAT* and (**f**) *JERF 3* genes in the leaf and root tissues of treated tomato plants compared to those in the untreated control (n = 6). (*) Symbol indicates a mean relative fold-change that is significantly different from that of the UC. Error bars indicate standard error reflecting variability within a sample (*p*-value ≤ 0.05). L refers to leaf tissues, and R refers to the root tissues, dpt refers to days post treatment.

*PR-3* gene expression was found to be high in the root tissues of the tomato plants treated with ACS5075. The highest expression was noted in the root tissues at 15 dpt, and this was 8-fold higher than that of the root tissues that were harvested at 3 dpt (Figure 1b). A similar trend of expression was also noted in roots treated with ACS3048 (Figure 2b), showing a 10-fold increase in *PR-3* gene expression at 15 dpt when compared to that at 3 dpt. However, *PR-3* gene expression in the leaf tissues of tomato plants treated with both products was downregulated, though it was not significantly different than that of the untreated control at most time points (Figure 1b and Figure 2b).

The *PR-5* gene expression was found to be significantly higher than that of the UC in most of the leaf and root tissues treated with both products (Figure 1c and Figure 2c). Its expression was high in both leaf and root tissues at 9 dpt when compared to that at all other days. The *PR-5* gene expression levels significantly dropped at 15 dpt in both leaf and root tissues in comparison to those at 3, 5, 9 dpt when treated with ACS5075 (Figure 1c). However, the treatment of plants with ACS3048 enhanced the *PR-5* gene expression in the root tissues over time. The 15 dpt root tissues had the highest *PR-5* gene expression, which was 2- and 1.4-fold higher compared to that at 3 and 9 dpt, respectively (Figure 2c).

ACS products increased *ACO* gene expression in both root and leaf tissues during the post-treatment days (Figure 1d and Figure 2d). Its expression was noted to be significantly high in ACS5075-treated leaf tissues at 5, 9 and 15 dpt, and it was 9-fold higher compared to that in the leaf tissues after 3 dpt (Figure 1d). A similar trend was also noted for the root tissues (Figure 1d) where the relative fold-change compared to untreated control tomato plants was 2-, 2.3- and 2.7-fold higher after 5, 9, and 15 dpt compared to 3 dpt. Treatment with ACS3048, showed a slightly different pattern of *ACO* gene expression in leaf tissues; its expression at 5, 9 and 15 dpt was significantly higher compared to that at 3 dpt, however, the expression levels tended to decrease from 5, 9 and 15 dpt. The *ACO* gene expression was found to be similar in root tissues of the tomato plants at 3 and 5 dpt. However, a 1.7-fold increase was noted in root tissue at 9 dpt; this increase was constant even at 15 dpt (Figure 2d).

The *CAT* gene was noted to be highly expressed in leaf tissues of the tomato plants treated with ACS5075 at 15 dpt when compared to that at 5 dpt (2-fold) and 9 dpt (1.6-fold) (Figure 1e). However, the *CAT* gene expression was not significantly different from that in the UC in the ACS5075-treated root tissues. Similarly, with ACS3048, leaf tissues recorded the highest *CAT* gene expression at 15 dpt compared to 3 dpt (25-fold), 5 dpt (10-fold) and 9 dpt (1.3-fold) (Figure 2e).

The *JERF 3* gene expression under the ACS5075 treatment was significantly different from that in the UC only in the leaf tissues at 9 and 15 dpt. Its expression in root tissue was significantly lower than that of the UC (Figure 1f). ACS3048 treatment had a significant impact on *JERF 3* gene expression in the treated leaf tissues at all time points examined. Its expression was highest at 5 dpt with a 3.5-fold change compared to that in the UC, followed by 9 dpt with a 3.4-fold change and 15 dpt had a 3-fold change compared to that in the UC (Figure 2f). The expression of the *JERF 3* gene was noted to be significantly downregulated at 3 dpt and 5 dpt in tomato root tissues treated with ACS3048 (Figure 2f).

Figure 3 represents fold-changes in expression of each defence gene (*PR-1*, *PR-3*, *PR-5*, *ACO*, *CAT* and *JERF 3*) in the root and leaf tissues of the tomato plants in the untreated + uninoculated control (UC), inoculated untreated (IU) and treated + inoculated test groups (T), against the housekeeping gene actin. This set of experiments had two controls i.e., untreated + uninoculated control (UC) and inoculated untreated control (IU); therefore, the y-axis in these graphs (Figure 3) represents the relative expression values of each gene compared to actin. In these graphs (Figure 3), an asterisk (*) indicates that the mean fold-change is significantly different from that of UC.

**Figure 3 microorganisms-11-01700-f003:**
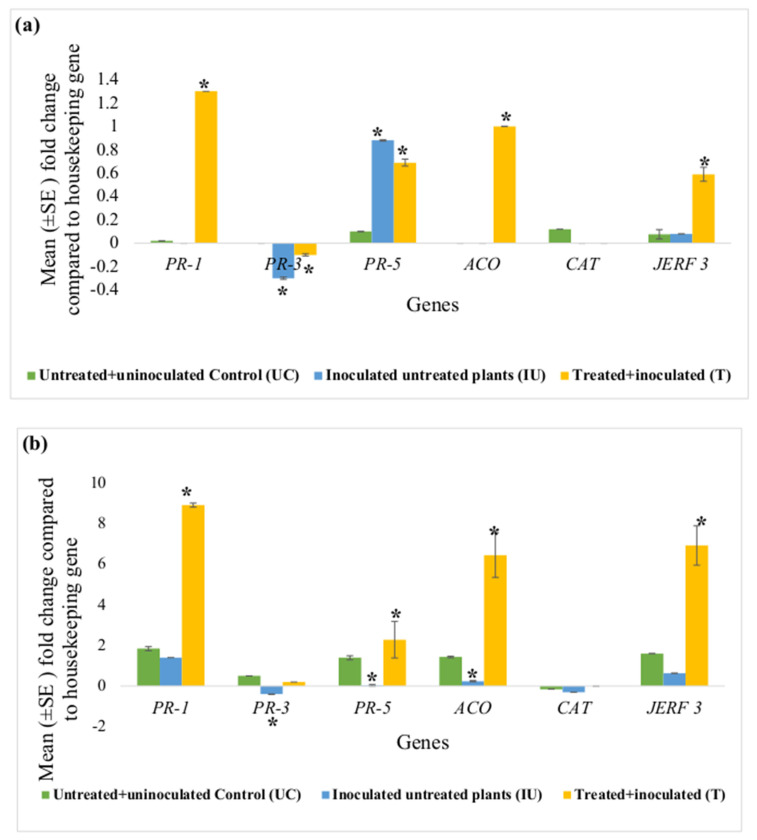
Effect of concurrent ACS5075 treatment and *M. javanica* inoculation after 15 dpi on (**a**) leaf tissues and (**b**) root tissues. Error bars indicate standard error, reflecting variability within a sample (n = 6). (*) Symbol indicates a mean fold-change that is significantly different from that of untreated+ uninoculated control (UC).

Comparing the UC and IU control plants, it was observed that in the leaf tissues, expression levels of none of the genes except for *PR-3* and *PR-5* were significantly different (Figure 3a). *PR-3* expression levels were noted to be significantly downregulated in the leaf tissues of IU plants compared to those in UC plants. However, its expression was still downregulated in the leaf tissues even after the treatment with ACS5075. *PR-5* gene expression levels significantly increased by 8.8-fold in the IU plants when compared to that in UC plants. This expression level did not change compared to that in IU plants in the leaf tissues after ACS5075 treatment. Upon treatment with ACS5075 (T), in the leaf tissues, *PR-1*, *JERF 3* and *ACO* expression levels were significantly higher compared to those in both UC and IU plants (Figure 3a). 

In the root tissue, most of the gene expression levels were reduced in the IU plants compared to those in the UC plants. However, with the treatment of ACS5075 (T), expression levels of *PR-1*, *PR-5*, *ACO* and *JERF 3* in the root tissue were significantly increased when compared to those in IU and UC plants (Figure 3b). The expression of *PR-1*, *PR-5*, *ACO* and *JERF 3* genes was 6.4-, 57-, 26.9- and 11-fold higher, respectively, in treated root tissues compared to that in the root tissues of the IU plants (Figure 3b).

Post-infection, in the ACS3048 treatment group, no significant fold-changes were noted for any of the genes analysed; therefore, the data are not presented.

### 3.2. Protein Analysis and Identification of Unique Proteins Using 2D Electrophoresis and N-Terminal Sequencing

Tomato plants were harvested after 9 days of treatment with the respective ACS products. Nine-day treatment was chosen, as most of the gene expression increased significantly after 9 days of treatment with the ACS products. Approximately 5 mg of protein was loaded in each well of an SDS-PAGE gel to determine the quality of the protein extracted from both the roots and leaves of treated plants. The presence of clear protein bands on the SDS-PAGE gel indicated that good quality protein was extracted from the plant tissue (Figure 4). Protein was less abundant in the samples extracted from the roots, as the respective bands on the SDS-PAGE gel appeared quite faint and diffuse (Lane CR, 75R, 48R and NG(R)) (Figure 4). Protein extraction from the root samples was difficult, and even after repeated extractions, the protein quality was consistently low; hence, bands both on SDS-PAGE and 2D-gels were not clearly distinguishable (Lane CR, 75R, 48R, NG(R)). 

All the protein samples extracted from the leaf tissues were analysed with 2D-gel electrophoresis to identify unique protein spots (Figure 5). The gels were run 10–12 times to ensure the presence of these spots. Protein spots 1, 2 and 3 were present in all samples (Figure 5a–d). 

A unique protein spot 4 was noted (Figure 5d) at 50 KDa in samples extracted from the plants treated with ACS3048, which was not observed with any other treatments, including untreated control samples. 

Upon analysis of the peptide sequences from the N-terminal sequencing, which had been carried out on each of the excised spots, the following proteins were identified: protein spot 1 was identified as ‘disease resistance protein RPP13-like’, protein spot 2 as ‘phosphatidylinositol 4-phosphate 5-kinase 2’, spot 3 as ‘protein SABRE like’ and protein spot 4 as ‘uncharacterized protein LOC101250254’ (https://blast.ncbi.nlm.nih.gov/Blast.cgi?CMD=Web&PAGE_TYPE=BlastHome) (accessed date: 20 April 2022). All the details related to the identified proteins are listed in Table 2.

**Figure 5 microorganisms-11-01700-f005:**
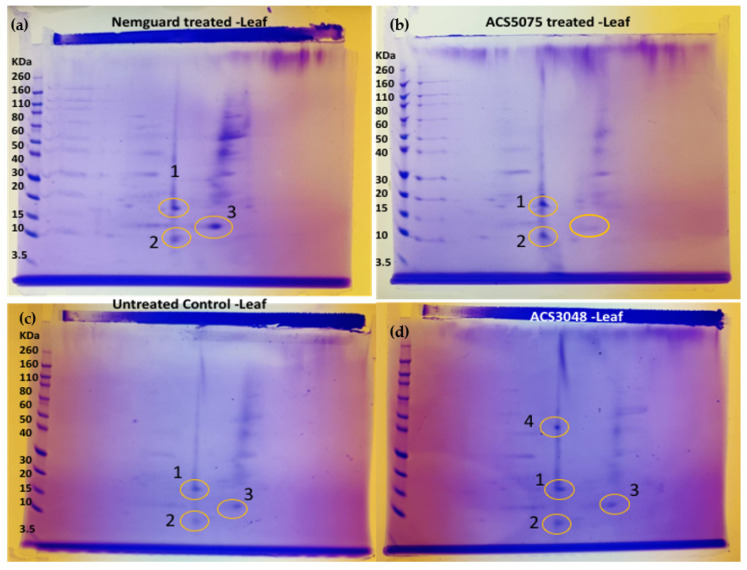
2D-gel images of the protein extracted from leaf tissues of tomato plants treated with (**a**) commercial organic nematicide Nemguard^TM^, (**b**) ACS5075, (**c**) untreated control and (**d**) ACS3048. The yellow circles indicate the protein spots. Numbers 1, 2, 3 and 4 indicate the protein spots.

## 4. Discussion

The treatment of tomato plants with ACS products has been reported to reduce root-knot nematode infestation and to also enhance tomato plant growth [16,17]. In this study, efforts have been made to understand the mode of action of these products and to explore if the products make plants more resistant to RKN infection by priming their defence processes. In tomato, the *JERF 3* gene (Acc. n. NM 001247533.2) encodes a trans-acting factor that responds to both ET and JA [12]. The ACC oxidase enzyme, which catalyses the final step of ET biosynthesis, is encoded by the *ACO* gene (Acc. n. XM 015225653.2) [22], while the catalase enzyme, encoded by the *CAT* gene (Acc. n. NM 001247257.2) [12], neutralises the harmful hydrogen peroxides produced by plants as a form of defence against pathogens and parasites.

In the current study tomato plants were treated with two ACS products, with and without RKN infection. Results indicate that treatment with both ACS products, ACS5075 and ACS3048, systemically primed *ACO* gene expression, as it was upregulated in both root and leaf tissues post-treatment and post-inoculation (Figure 1, Figure 2 and Figure 3). Similar results were reported in tomato plants treated with a mixture of beneficial biocontrol agents (BCA; [12]). In the primed state, plants have stronger and faster defence against pathogen attack due to their activated immune system. Upregulation of the *ACO* gene could enhance ET levels in the primed root and leaf tissues; ethylene is known to play an important role in plant defence against endoparasitic sedentary nematodes [23]. 

Apart from their contribution towards plant growth and development, plant hormones also play a very important role in enhancing plant responses to biotic and abiotic stresses [24,25,26]. Enhanced ET and ET signalling could increase hormone signalling crosstalk and therefore could also enhance the plant response to nematode infestation and subsequently induce resistance against RKN infection [23]. The ET signalling pathway is also reported to reduce attraction of soybean cyst nematodes (*Heterodera glycines*) towards *Arabidopsis* roots [23].

Among the PR genes, the *PR-1* gene was found to be upregulated in both root and leaf tissues of tomato plants treated with ACS5075 post-infection with RKN and with ACS3048 post treatment without RKN infection (Figure 2 and Figure 3). Similar upregulation of the *PR-1* gene was reported in wheat upon treatment with a fermentation-based elicitor, which further enhanced protection against powdery mildew in wheat [18]. *PR-1* gene expression has been used as a useful molecular marker of SAR-mediated disease resistance [27]. The *PR-1* family members have been reported to bind sterols [27] and are known to manifest antimicrobial activity towards sterol auxotrophs. The *PR-1* gene is also reported to be induced upon treatment of tomato plants with salicylic acid and to therefore protect plants against biotrophic pathogens [12]. Higher *PR-1* gene expression is also known to trigger resistance in tobacco cultivars primed with the fungicides strobilurin and pyraclostrobin against tobacco mosaic virus and *Pseudomonas syringae pv. tabaci* infection [28]. Enhanced *PR-1* gene expression levels are known to enhance the of activity MPK3/6 proteins, which are known to play a very important role in regulating multiple defence pathways in plant fungal resistance [29]. Upregulation of the *PR-1* gene is also reported to trigger immunity in tomato plants against *P. syringae* [30] and RKN [12] infection. Post-inoculation with *M. javanica* and treatment with ACS5075, in the root tissues, a positive correlation (r = 0.4 and 0.2) has been noted while correlating the expression of *PR-1* and *ACO* genes, *PR-5* and *JERF 3* genes, respectively. This means with an increase in the expression of one gene, there is a positive increase in the fixed proportion of the expression of other genes. In the leaf tissues, a strong positive correlation (r = 0.8, 0.7, 0.6) was also noted while correlating *ACO* and *JERF 3*, *JERF 3* and *PR-1* and *ACO* and *PR-1* genes, respectively. Overall, the expression of *ACO* and *PR-1* genes was noted to increase in both root and leaf tissues thus playing an important role in enhancing plant defence against nematode infestation.

Changes in mRNA expression have biological meaning, most likely mediated by corresponding changes in protein levels, though this is not always the case. Protein extracts from the ACS-treated plants were examined to possibly confirm that the gene expression analysis translated to protein changes. Protein spot 1 was identified in all the tomato plants treated with ACS products and Nemguard^TM^ (Figure 5). Upon sequencing, this protein was identified to be ‘disease resistance protein RPP13-like’. This protein is known to have a very important function in conferring resistance to five isolates of *Peronospora parasitica*, which is a fungus reported to cause downy mildew in plants [31,32]. This protein belongs to a class of R- proteins, some of which (RPS2, RPM1 and RPS5) are known to confer resistance against bacteria [33], nematodes (*Mi-1.2*; [34]) and fungi (*RPP8-Ler* and *I2*; [35]). The presence of this protein could be one of the reasons for reduced *M. javanica* infestation in ACS-treated tomato plants. The occurrence of this ‘disease resistance protein RPP13-like’ protein could also be due to the higher expression of the *PR-1* gene [29]. However, this protein spot was also observed even in the untreated control plants. Detailed future analysis is required to fully understand this observation.

Protein spot 2 (Figure 5) has been identified as ‘phosphatidylinositol 4-phosphate 5-kinase 2’. This protein belongs to a class of proteins that phosphorylate phosphatidylinositol 4-phosphate. These proteins have been previously reported to control membrane trafficking and to contribute towards the growth and development of *Arabidopsis* spp. [36]. They are also functionally linked to the regulation of numerous physiological processes in plants, such as membrane trafficking, clathrin-mediated endocytosis and the dynamics of the actin cytoskeleton [36,37,38] and may also be associated with the increased leaf biomass of the treated tomato plants, which was observed in a previous study [17]. Protein spot 3 (Figure 5) was identified as ‘protein SABRE-like’ (Table 2), and this protein was found in the plants treated with Nemguard^TM^ and ACS3048 (Figure 5). This protein is reported to enhance pollen tubes, root tip and root hair growth and cell expansion in *Arabidopsis* spp. and other eukaryotes [26]; indeed, an increase in root length was observed in potatoes treated with both ACS products [39]. This protein is known to counter balance the activity of ET to avoid abnormal cell expansion in *Arabidopsis* species [40]. Therefore, SABRE-protein activity is essential for the plant cells to attain a normal shape by actively contributing towards attaining ‘dynamic equilibrium’ [40]. 

Results indicate that treatment of tomato plants with both Alltech products, ACS5075 and ACS3048, primed *ACO* gene expression, as it was upregulated in both root and leaf tissues post-treatment and post-inoculation. In the primed state, plants have stronger and faster defence against pathogen attack due to their activated immune system [11]. Similarly with the PR genes, the *PR-1* gene was upregulated in both root and leaf tissues of tomato plants treated with ACS5075 post-infection with RKN and with ACS3048 post-treatment without RKN infection. Upregulation of the *PR-1* gene is known to trigger immunity in tomato plants facing pathogenic challenges [12]. These results show that the ACS products used in this study activated immune responses in the plants prior to challenge, and this response continued following challenge with RKN. This indicates the potential for these products as immune response-elicitors to help protect plants from biotic challenges. Although some protein differences were noted in this work, a more complete proteomic analysis is required to confirm the protein defence-response in the ACS-treated plants.

## Figures and Tables

**Figure 4 microorganisms-11-01700-f004:**
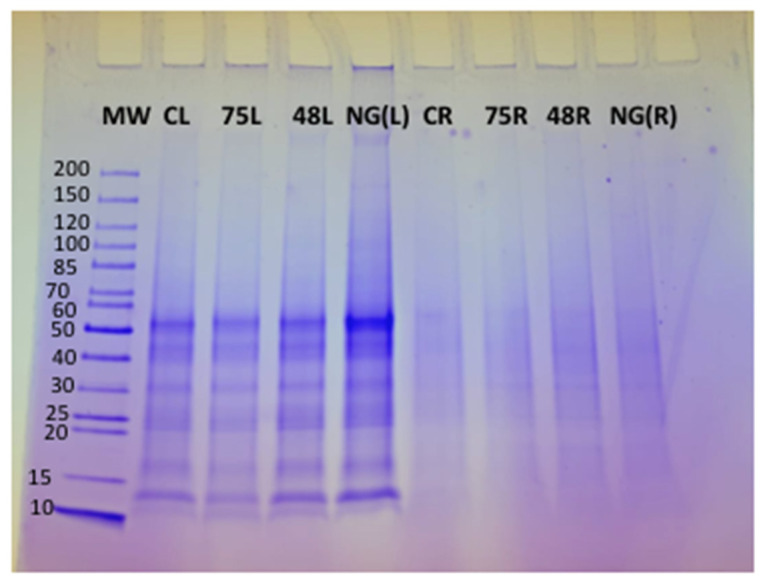
SDS-PAGE gel of protein extracted from tomato plants treated with ACS products and Nemguard^TM^. Lane 1: Pre-stained molecular weight protein ladder (10–200 KDa) (MW), Lane 2: untreated control leaf tissues (CL), Lane 3: ACS5075-treated leaf tissues (75L), Lane 4: ACS3048-treated leaf tissues (48L), Lane 5: Nemguard^TM^ treated leaf tissues NG(L), Lane 6: untreated control root tissues (CR), Lane 7: ACS5075-treated root tissues (75R), Lane 8: ACS3048-treated root tissues (48R) and Lane 9: Nemguard^TM^-treated root tissues NG(R).

**Table 1 microorganisms-11-01700-t001:** List of defence-related genes studied and the specific primers used in reverse transcriptase- real time polymerase chain reaction (RT-PCR).

Gene	GenBank Accession Number	Function	Primer Sequence (F; Forward and R; Reverse)	References
*PR-1*	AF384143.1	Pathogenesis-related 1, β-1,3-glucanases Involved in stress response and plant defence and play a role in the regulation of callose deposition and in hydrolysis of the fungal cell wall	F: CAATAACCTCGGCGTCTTCATCAC R: TTATTTACTCGCTCGGTCCCTCTG	[18]
*PR-3*	NM_001247474.2	Chitinase Encodes several types of endochitinases and has generally been reported to be induced by activation of the JA-signalling pathway and ethylene treatments in tomato	F: AACTATGGGCCATGTGGAAGA R: GGCTTTGGGGATTGAGGAG	[12]
*PR-5*	NM_001247422.3	Pathogenesis-related 5 Encodes thaumatin-like proteins and is involved in osmotic regulation of cells	F: GCAACAACTGTCCATACACC R: AGACTCCACCACAATCACC	[12]
*JERF 3*	NM_001247533.2	Jasmonate ethylene response factor Member of ERF proteins, a trans-acting factor responding to both ET and JA in tomato	F: GCCATTTGCCTTCTCTGCTTC R: GCAGCAGCATCCTTGTCTGA	[12]
*ACO*	XM_015225653.2	1-Aminocyclopropane-1-carboxylic acidoxidase Enzyme catalyses the last step of ethylene biosynthesis	F: CCATCATTTCTCCAGCATCA R: TTGGCAGACTCAAATCTAGG	[12]
*CAT*	NM_001247257.2	Catalase 2 Neutralizes the toxic hydrogen peroxides produced in plant defence against pathogens and parasites	F: TGCTCCAAAGTGTGCTCATC R: TTGCATCCTCCTCTGAAACC	[12]
*Actin*	NM_001321306.1	Actin-7-like Housekeeping gene	F: GATACCTGCAGCTTCCATACC R: GCTTTGCCGCATGCCATTCT	[12]

**Table 2 microorganisms-11-01700-t002:** List of specific proteins identified from the individual protein spots.

Protein Spots	Sequence	Name of Protein	NCBI Reference Sequence	Percent Identity	Reference
Protein spot 1	EVRSF	disease resistance protein RPP13-like	XP_010319326.1	100%	NCBI|PBLAST
Protein spot 2	FQVDP	phosphatidylinositol 4-phosphate 5-kinase 2	XP_004250336.1	100%	NCBI|PBLAST
Protein spot 3	VEPA	protein SABRE-like	XP_015060204.1	100%	NCBI|PBLAST
Protein spot 4	KVMPFEA	uncharacterized protein LOC101250254	XP_004230480.1	100%	NCBI|PBLAST

## Data Availability

Data have been stored in the library repository of South East Technological University, Carlow, Ireland.

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
