# Peer review of "Investigating the Effects of Alltech Crop Science (ACS) Products on Plant Defence against Root-Knot Nematode Infestation"

_microorganisms, 2023, doi:10.3390/microorganisms11071700_

Round 1
Reviewer 1 Report (Previous Reviewer 1)
Dear authors,
The manuscript “Investigating Effects of Alltech Crop Science (ACS) Products on Plant Defence Mechanisms following Root‐Knot Nematode Infestation” has been improved with the revisions of the reviewers. Some minor revisions are proposed to improve the quality of the manuscript below.
Line 34. Elsen {4} . Those brackets are in journal format?
Line 38. Would you change Meloidogyne to M. ?
Lines 51 to 54. Need a reference
Line 66. I would write “among others” because not only Rhizobacterias are responsible for that activation
Lines 94. Solanum lycopersicum should be in italics
Line 94 and 109 Delete variety
Line 96. What is a SETU?
Line 118, I understand, in the solid treatment, you remove the plant, mix it with the product and then put the plant back into the pot?
Detailed product treatments should be explained
Line 121. Change days post-treatment by dpt
Lone 120-122. What changes between 3dpt and 5dpt and all the other dpt? If it is the same, rewrite the sentence. Proposal “Plants were uprooted after each dpi to proceed with RNA extraction and then RT‐PCR” and then lines 120 – 126 I would remove it.
Line 129. I would remove (100mg per sample) and explain better the complete sentence. Proposal: “Leaf and root tissue, separately, was ground to a fine powder using liquid nitrogen with a sterile porcelain pestle and mortar, and 100mg per sample was used for RNA extraction immediately or kept at -80ºC until it's used.”
Line 434 and Table 1. Molinari 21, studied JERF3. Is it the same as you study? If it is the same I would change all JERF to JERF3.
Line 456. Change soyabean cyst nematodes to soybean cyst nematodes
Ine477. Change possibly to possible
Table 1. Line 219. I propose to include a column with the references and not in the footnote
Figure 1. needs to be improved.
Figure 2. needs to be improved.
Figure 3 needs to be improved. Foot notes look blurry
Author Response
Response to reviewer comments
|
Reviewer 1 |
|
||||||||||||||||||||||||||||||||||||||||||||||||||||||||||||||||||||||||
|
The manuscript “Investigating Effects of Alltech Crop Science (ACS) Products on Plant Defence Mechanisms following Root‐Knot Nematode Infestation” has been improved with the revisions of the reviewers. Some minor revisions are proposed to improve the quality of the manuscript below. |
|
||||||||||||||||||||||||||||||||||||||||||||||||||||||||||||||||||||||||
|
Line 34. Elsen {4} . Those brackets are in journal format? |
Corrected to square brackets |
||||||||||||||||||||||||||||||||||||||||||||||||||||||||||||||||||||||||
|
Line 38. Would you change Meloidogyne to M. ? |
Changed |
||||||||||||||||||||||||||||||||||||||||||||||||||||||||||||||||||||||||
|
Lines 51 to 54. Need a reference |
Added |
||||||||||||||||||||||||||||||||||||||||||||||||||||||||||||||||||||||||
|
Line 66. I would write “among others” because not only Rhizobacterias are responsible for that activation |
Edited accordingly |
||||||||||||||||||||||||||||||||||||||||||||||||||||||||||||||||||||||||
|
Lines 94. Solanum lycopersicum should be in italics |
Changed |
||||||||||||||||||||||||||||||||||||||||||||||||||||||||||||||||||||||||
|
Line 94 and 109 Delete variety |
Deleted |
||||||||||||||||||||||||||||||||||||||||||||||||||||||||||||||||||||||||
|
Line 121. Change days post-treatment by dpt |
Changed |
||||||||||||||||||||||||||||||||||||||||||||||||||||||||||||||||||||||||
|
Line 96. What is a SETU? |
Its South East Technological University (SETU) |
||||||||||||||||||||||||||||||||||||||||||||||||||||||||||||||||||||||||
|
Line 118, I understand, in the solid treatment, you remove the plant, mix it with the product and then put the plant back into the pot?
Detailed product treatments should be explained |
Detailed explanation about product treatments have been written within this paragraph. |
||||||||||||||||||||||||||||||||||||||||||||||||||||||||||||||||||||||||
|
Lone 120-122. What changes between 3dpt and 5dpt and all the other dpt? If it is the same, rewrite the sentence. Proposal “Plants were uprooted after each dpi to proceed with RNA extraction and then RT‐PCR” and then lines 120 – 126 I would remove it. |
Edited accordingly |
||||||||||||||||||||||||||||||||||||||||||||||||||||||||||||||||||||||||
|
Line 129. I would remove (100mg per sample) and explain better the complete sentence. Proposal: “Leaf and root tissue, separately, was ground to a fine powder using liquid nitrogen with a sterile porcelain pestle and mortar, and 100mg per sample was used for RNA extraction immediately or kept at -80ºC until it's used.” |
Edited |
||||||||||||||||||||||||||||||||||||||||||||||||||||||||||||||||||||||||
|
Line 434 and Table 1. Molinari 21, studied JERF3. Is it the same as you study? If it is the same I would change all JERF to JERF3. |
Have been changed |
||||||||||||||||||||||||||||||||||||||||||||||||||||||||||||||||||||||||
|
Line 456. Change soyabean cyst nematodes to soybean cyst nematodes |
changed |
||||||||||||||||||||||||||||||||||||||||||||||||||||||||||||||||||||||||
|
Ine477. Change possibly to possible |
changed |
||||||||||||||||||||||||||||||||||||||||||||||||||||||||||||||||||||||||
|
Table 1. Line 219. I propose to include a column with the references and not in the footnote |
A column with references have been included and footnote deleted |
||||||||||||||||||||||||||||||||||||||||||||||||||||||||||||||||||||||||
|
Figure 1. needs to be improved. Figure 2. needs to be improved.
Figure 3 needs to be improved. Foot notes look blurry
|
Both Figure 1 and 2 have been improved and have been uploaded as a separate word file named “Figure 1, 2 and 3”
Figure 3 has been improved and the clarity of foot notes has been improved.
All Figure 1, 2 and 3 have been improved and have been uploaded as a separate word file named “Figure 1, 2 and 3” |
||||||||||||||||||||||||||||||||||||||||||||||||||||||||||||||||||||||||
|
Reviewer 3 |
|
||||||||||||||||||||||||||||||||||||||||||||||||||||||||||||||||||||||||
|
The authors need to revise the title of the paper in a more meaningful way. the title is long and has some unnecessary information, suggestion: “Investigating the Effects of Alltech Crop Science Products on Plant Defense Against Root-Knot Nematodes.”
|
Title has been changed as per the suggestion |
||||||||||||||||||||||||||||||||||||||||||||||||||||||||||||||||||||||||
|
The abstract is written in a way lacks logic. It should highlight the salient findings more critically; |
Abstract has been edited |
||||||||||||||||||||||||||||||||||||||||||||||||||||||||||||||||||||||||
|
|
|
||||||||||||||||||||||||||||||||||||||||||||||||||||||||||||||||||||||||
|
In the materials and methods of the manuscript, in the section where the RT-PCR procedures are described and in the statistical analyses, the authors report that they performed a relative expression using the Livak 2001 method; However, it was not clear in the text who was the calibrating sample used in the relationship parameters. TO DESCRIBE!!!
|
The cycle threshold (CT values) that were generated by the Roche machine were recorded individually. ΔCT values were calculated by deducting the individual CT values from that of the CT values obtained for the respective Actin gene per each sample [ΔCT(a target sample)−ΔCT(Actin)]. Using a reference gene (Actin) as a standard, the ultimate outcome of this method is represented as the mean relative fold change in target gene expression in a target sample compared to untreated control, which acted as a calibrating sample.
The above description has been added in section 2.3 under materials and methods. |
||||||||||||||||||||||||||||||||||||||||||||||||||||||||||||||||||||||||
|
Authors should discuss the results integrally. The discussion is based on individual results. I suggest that integrating the results will give more value to the work. I suggest that you discuss by integrating all your results. You can use correlation tests (PCA or Pearson Correlation).
|
Post-inoculation with M. javanica and treatment with ACS5075, in the root tissues, a positive correlation (r=0.4 and 0.2) has been noted while correlating expression of PR-1 and ACO genes, PR-5 and JERF 3 genes, respectively. This means with an increase in expression of one gene there is a positive increase of fixed proportion in the expression of other genes.
In the leaf tissues, a strong positive correlation (r=0.8, 0.7, 0.6) was also noted while correlating ACO and JERF 3, JERF 3 and PR-1, ACO and PR-1 genes, respectively.
All of the above has been added under the discussion section. |
||||||||||||||||||||||||||||||||||||||||||||||||||||||||||||||||||||||||
|
The conclusion is totally confusing. Re-write the conclusion! It needs to be much improved. |
Conclusion part has been re-written |
||||||||||||||||||||||||||||||||||||||||||||||||||||||||||||||||||||||||
|
Reviewer 2 |
|
||||||||||||||||||||||||||||||||||||||||||||||||||||||||||||||||||||||||
|
Introduction Line 53 Please add a reference concerning the other approaches used to control PPN |
Reference has been added |
||||||||||||||||||||||||||||||||||||||||||||||||||||||||||||||||||||||||
|
Material and Methods 2.4 How many plants did you use per treatment? Please indicate the number |
All the treatments were conducted in triplicates. It has been mentioned in section 2.4. |
||||||||||||||||||||||||||||||||||||||||||||||||||||||||||||||||||||||||
|
Results Lines 237-242 Expression of gene PR1 was referred to the untreated control. I understand that the value of UC is 1, but no reference is inserted in the text or in the legends of the Figures 1 and 2. I recommend adding a small sentence i.e. "compared to untreated control (set as a unit) at all times" in the legends of figures 1 and 2 or a dotted line at value 1 level, in all graphs, which indicates the expression of the untreated control. |
The y-axis in figure 1 and 2 indicates the mean fold change compared to UC. It’s a ratio. If the ratio was equal to 1, it means the expression was not different from that of UC. If the ratio was more or less than 1, it indicates that the expression was different from that of UC. All of this explanation has been included under ‘statistical analysis’ section. |
||||||||||||||||||||||||||||||||||||||||||||||||||||||||||||||||||||||||
|
Lines 248-249 You are comparing expression in roots between 15dpt and 3dpt. According to the graph 1b the fold change is more than 4-fold higher because the value of 3dpt is set at 0.5. Please recalculate.
|
The values have been recalculated and corrected accordingly |
||||||||||||||||||||||||||||||||||||||||||||||||||||||||||||||||||||||||
|
Line 251 Also in this case the value is not correct. Please check and recalculate |
The values have been recalculated and corrected accordingly |
||||||||||||||||||||||||||||||||||||||||||||||||||||||||||||||||||||||||
|
Figure 1 Please uniform all graphs, some have grid others no.
|
All the grids have been removed to make them look uniform. |
||||||||||||||||||||||||||||||||||||||||||||||||||||||||||||||||||||||||
|
Line 282 You can state significantly lower if there are asterisks.
|
(*) Symbol indicates mean relative fold change which is significantly different from that of the UC. This sentence has been indicated in the legends of figure 1, 2 and 3. |
||||||||||||||||||||||||||||||||||||||||||||||||||||||||||||||||||||||||
|
Line 311 It seems to me that there are mistakes in the calculation of folds of expression in treated tissues respect to IU. Please check and recalculate.
|
The fold of expression have been re-calculated and corrected accordingly within the text |
||||||||||||||||||||||||||||||||||||||||||||||||||||||||||||||||||||||||
|
Figure 5 shows that protein spots 1, 2 and 3 were present in samples extracted from Nemguard, ACS3048 treated plants and untreated control. For corroborating your statement “However, these protein spots were faintly observed even in the untreated control plants” I suggest to add a graph showing the quantitation of the expressed bands
|
Figure 5 shows that protein spots 1,2and 3 were present in all the as indicated with the yellow circles.
We have not performed any gel analysis or densitometry, therefore, quantitation of the expressed bands is not possible at this stage. This will be kept in mind for all our future studies. I have edited the sentence in the manuscript as “However, these protein spots were also observed even in the untreated control plants”.
|
||||||||||||||||||||||||||||||||||||||||||||||||||||||||||||||||||||||||
|
Minor points: Introduction Line 38 Because you already cited the genus Meloidogyne (i.e. Meloidogyne arenaria) you can shorten M. incognita, M. javanica and so on.
|
edited |
||||||||||||||||||||||||||||||||||||||||||||||||||||||||||||||||||||||||
|
Material and Methods 2.1 Line 94 Solanum lycopersicum shoud be in Italic |
edited |
||||||||||||||||||||||||||||||||||||||||||||||||||||||||||||||||||||||||
|
Line 94 Delete “variety”
|
deleted |
||||||||||||||||||||||||||||||||||||||||||||||||||||||||||||||||||||||||
|
Line 95 Better “roots that were infected with M. javanica”
|
changed |
||||||||||||||||||||||||||||||||||||||||||||||||||||||||||||||||||||||||
|
2.2 Line 109 Delete “variety”
|
deleted |
||||||||||||||||||||||||||||||||||||||||||||||||||||||||||||||||||||||||
|
Lines 117-119 I suggest to rewrite the sentence as “…..4-week old tomato plants were treated individually with 3% (v/v) ACS5075 or 3g ACS3048 only once and were harvested after 3, 5, 9 and 15 days”
|
rewritten |
||||||||||||||||||||||||||||||||||||||||||||||||||||||||||||||||||||||||
|
Lines 119-123 I suggest to move this part “Six replications….including the control” immediately after line 126 “…RKN study[17]”
|
edited accordingly |
||||||||||||||||||||||||||||||||||||||||||||||||||||||||||||||||||||||||
|
Line 126 Add a space between RKN and study
|
added |
||||||||||||||||||||||||||||||||||||||||||||||||||||||||||||||||||||||||
|
Line 127 I suggest to delete “After the completion of each treatment…from the soil” because it is a repetition
|
deleted |
||||||||||||||||||||||||||||||||||||||||||||||||||||||||||||||||||||||||
|
Lines 131-132 Delete “All the treatments were conducted with six replications, therefore”
|
deleted |
||||||||||||||||||||||||||||||||||||||||||||||||||||||||||||||||||||||||
|
Line 132 Twelve (now capital letter)
|
changed |
||||||||||||||||||||||||||||||||||||||||||||||||||||||||||||||||||||||||
|
2.5 Line 215 S. lycopersicum
|
changed |
||||||||||||||||||||||||||||||||||||||||||||||||||||||||||||||||||||||||
|
Results Line 250 Better to write “trend” instead of “pattern”
|
Replaced ‘pattern’ with ‘trend’ |
||||||||||||||||||||||||||||||||||||||||||||||||||||||||||||||||||||||||
|
Line 286 This sentence is incomplete
|
edited |
||||||||||||||||||||||||||||||||||||||||||||||||||||||||||||||||||||||||
|
Figure 3 “housekeeping” not “house keeping”
|
changed |
||||||||||||||||||||||||||||||||||||||||||||||||||||||||||||||||||||||||
|
Discussion Lines 446-448 Delete this sentence. It has already written. |
deleted |
||||||||||||||||||||||||||||||||||||||||||||||||||||||||||||||||||||||||
|
Line 456 soybean
|
changed |
||||||||||||||||||||||||||||||||||||||||||||||||||||||||||||||||||||||||
|
Line 473 P. syringae
|
corrected |
||||||||||||||||||||||||||||||||||||||||||||||||||||||||||||||||||||||||
|
Lines 480 and 491 Delete Table 2
|
Deleted |
||||||||||||||||||||||||||||||||||||||||||||||||||||||||||||||||||||||||
|
|
|
||||||||||||||||||||||||||||||||||||||||||||||||||||||||||||||||||||||||

Reviewer 2 Report (Previous Reviewer 2)
To the Authors,
In this revised paper many suggestions by reviewers were addressed. However, there are still some flaws.
Introduction
Line 53 Please add a reference concerning the other approaches used to control PPN
Material and Methods
2.4
How many plants did you use per treatment? Please indicate the number
Results
Lines 237-242 Expression of gene PR1 was referred to the untreated control. I understand that the value of UC is 1, but no reference is inserted in the text or in the legends of the Figures 1 and 2. I recommend adding a small sentence i.e. "compared to untreated control (set as a unit) at all times" in the legends of figures 1 and 2 or a dotted line at value 1 level, in all graphs, which indicates the expression of the untreated control.
Lines 248-249 You are comparing expression in roots between 15dpt and 3dpt. According to the graph 1b the fold change is more than 4-fold higher because the value of 3dpt is set at 0.5. Please recalculate.
Line 251 Also in this case the value is not correct. Please check and recalculate
Figure 1 Please uniform all graphs, some have grid others no.
Line 282 You can state significantly lower if there are asterisks.
Line 311 It seems to me that there are mistakes in the calculation of folds of expression in treated tissues respect to IU. Please check and recalculate.
Figure 5 shows that protein spots 1, 2 and 3 were present in samples extracted from Nemguard, ACS3048 treated plants and untreated control. For corroborating your statement “However, these protein spots were faintly observed even in the untreated control plants” I suggest to add a graph showing the quantitation of the expressed bands
Minor points:
Introduction
Line 38 Because you already cited the genus Meloidogyne (i.e. Meloidogyne arenaria) you can shorten M. incognita, M. javanica and so on.
Material and Methods
2.1
Line 94 Solanum lycopersicum shoud be in Italic
Line 94 Delete “variety”
Line 95 Better “roots that were infected with M. javanica”
2.2
Line 109 Delete “variety”
Lines 117-119 I suggest to rewrite the sentence as “…..4-week old tomato plants were treated individually with 3% (v/v) ACS5075 or 3g ACS3048 only once and were harvested after 3, 5, 9 and 15 days”
Lines 119-123 I suggest to move this part “Six replications….including the control” immediately after line 126 “…RKN study[17]”
Line 126 Add a space between RKN and study
Line 127 I suggest to delete “After the completion of each treatment…from the soil” because it is a repetition
Lines 131-132 Delete “All the treatments were conducted with six replications, therefore”
Line 132 Twelve (now capital letter)
2.5
Line 215 S. lycopersicum
Results
Line 250 Better to write “trend” instead of “pattern”
Line 286 This sentence is incomplete
Figure 3 “housekeeping” not “house keeping”
Discussion
Lines 446-448 Delete this sentence. It has already written.
Line 456 soybean
Line 473 P. syringae
Lines 480 and 491 Delete Table 2
Author Response
Response to reviewer comments
|
Reviewer 1 |
|
||||||||||||||||||||||||||||||||||||||||||||||||||||||||||||||||||||||||
|
The manuscript “Investigating Effects of Alltech Crop Science (ACS) Products on Plant Defence Mechanisms following Root‐Knot Nematode Infestation” has been improved with the revisions of the reviewers. Some minor revisions are proposed to improve the quality of the manuscript below. |
|
||||||||||||||||||||||||||||||||||||||||||||||||||||||||||||||||||||||||
|
Line 34. Elsen {4} . Those brackets are in journal format? |
Corrected to square brackets |
||||||||||||||||||||||||||||||||||||||||||||||||||||||||||||||||||||||||
|
Line 38. Would you change Meloidogyne to M. ? |
Changed |
||||||||||||||||||||||||||||||||||||||||||||||||||||||||||||||||||||||||
|
Lines 51 to 54. Need a reference |
Added |
||||||||||||||||||||||||||||||||||||||||||||||||||||||||||||||||||||||||
|
Line 66. I would write “among others” because not only Rhizobacterias are responsible for that activation |
Edited accordingly |
||||||||||||||||||||||||||||||||||||||||||||||||||||||||||||||||||||||||
|
Lines 94. Solanum lycopersicum should be in italics |
Changed |
||||||||||||||||||||||||||||||||||||||||||||||||||||||||||||||||||||||||
|
Line 94 and 109 Delete variety |
Deleted |
||||||||||||||||||||||||||||||||||||||||||||||||||||||||||||||||||||||||
|
Line 121. Change days post-treatment by dpt |
Changed |
||||||||||||||||||||||||||||||||||||||||||||||||||||||||||||||||||||||||
|
Line 96. What is a SETU? |
Its South East Technological University (SETU) |
||||||||||||||||||||||||||||||||||||||||||||||||||||||||||||||||||||||||
|
Line 118, I understand, in the solid treatment, you remove the plant, mix it with the product and then put the plant back into the pot?
Detailed product treatments should be explained |
Detailed explanation about product treatments have been written within this paragraph. |
||||||||||||||||||||||||||||||||||||||||||||||||||||||||||||||||||||||||
|
Lone 120-122. What changes between 3dpt and 5dpt and all the other dpt? If it is the same, rewrite the sentence. Proposal “Plants were uprooted after each dpi to proceed with RNA extraction and then RT‐PCR” and then lines 120 – 126 I would remove it. |
Edited accordingly |
||||||||||||||||||||||||||||||||||||||||||||||||||||||||||||||||||||||||
|
Line 129. I would remove (100mg per sample) and explain better the complete sentence. Proposal: “Leaf and root tissue, separately, was ground to a fine powder using liquid nitrogen with a sterile porcelain pestle and mortar, and 100mg per sample was used for RNA extraction immediately or kept at -80ºC until it's used.” |
Edited |
||||||||||||||||||||||||||||||||||||||||||||||||||||||||||||||||||||||||
|
Line 434 and Table 1. Molinari 21, studied JERF3. Is it the same as you study? If it is the same I would change all JERF to JERF3. |
Have been changed |
||||||||||||||||||||||||||||||||||||||||||||||||||||||||||||||||||||||||
|
Line 456. Change soyabean cyst nematodes to soybean cyst nematodes |
changed |
||||||||||||||||||||||||||||||||||||||||||||||||||||||||||||||||||||||||
|
Ine477. Change possibly to possible |
changed |
||||||||||||||||||||||||||||||||||||||||||||||||||||||||||||||||||||||||
|
Table 1. Line 219. I propose to include a column with the references and not in the footnote |
A column with references have been included and footnote deleted |
||||||||||||||||||||||||||||||||||||||||||||||||||||||||||||||||||||||||
|
Figure 1. needs to be improved. Figure 2. needs to be improved.
Figure 3 needs to be improved. Foot notes look blurry
|
Both Figure 1 and 2 have been improved and have been uploaded as a separate word file named “Figure 1, 2 and 3”
Figure 3 has been improved and the clarity of foot notes has been improved.
All Figure 1, 2 and 3 have been improved and have been uploaded as a separate word file named “Figure 1, 2 and 3” |
||||||||||||||||||||||||||||||||||||||||||||||||||||||||||||||||||||||||
|
Reviewer 3 |
|
||||||||||||||||||||||||||||||||||||||||||||||||||||||||||||||||||||||||
|
The authors need to revise the title of the paper in a more meaningful way. the title is long and has some unnecessary information, suggestion: “Investigating the Effects of Alltech Crop Science Products on Plant Defense Against Root-Knot Nematodes.”
|
Title has been changed as per the suggestion |
||||||||||||||||||||||||||||||||||||||||||||||||||||||||||||||||||||||||
|
The abstract is written in a way lacks logic. It should highlight the salient findings more critically; |
Abstract has been edited |
||||||||||||||||||||||||||||||||||||||||||||||||||||||||||||||||||||||||
|
|
|
||||||||||||||||||||||||||||||||||||||||||||||||||||||||||||||||||||||||
|
In the materials and methods of the manuscript, in the section where the RT-PCR procedures are described and in the statistical analyses, the authors report that they performed a relative expression using the Livak 2001 method; However, it was not clear in the text who was the calibrating sample used in the relationship parameters. TO DESCRIBE!!!
|
The cycle threshold (CT values) that were generated by the Roche machine were recorded individually. ΔCT values were calculated by deducting the individual CT values from that of the CT values obtained for the respective Actin gene per each sample [ΔCT(a target sample)−ΔCT(Actin)]. Using a reference gene (Actin) as a standard, the ultimate outcome of this method is represented as the mean relative fold change in target gene expression in a target sample compared to untreated control, which acted as a calibrating sample.
The above description has been added in section 2.3 under materials and methods. |
||||||||||||||||||||||||||||||||||||||||||||||||||||||||||||||||||||||||
|
Authors should discuss the results integrally. The discussion is based on individual results. I suggest that integrating the results will give more value to the work. I suggest that you discuss by integrating all your results. You can use correlation tests (PCA or Pearson Correlation).
|
Post-inoculation with M. javanica and treatment with ACS5075, in the root tissues, a positive correlation (r=0.4 and 0.2) has been noted while correlating expression of PR-1 and ACO genes, PR-5 and JERF 3 genes, respectively. This means with an increase in expression of one gene there is a positive increase of fixed proportion in the expression of other genes.
In the leaf tissues, a strong positive correlation (r=0.8, 0.7, 0.6) was also noted while correlating ACO and JERF 3, JERF 3 and PR-1, ACO and PR-1 genes, respectively.
All of the above has been added under the discussion section. |
||||||||||||||||||||||||||||||||||||||||||||||||||||||||||||||||||||||||
|
The conclusion is totally confusing. Re-write the conclusion! It needs to be much improved. |
Conclusion part has been re-written |
||||||||||||||||||||||||||||||||||||||||||||||||||||||||||||||||||||||||
|
Reviewer 2 |
|
||||||||||||||||||||||||||||||||||||||||||||||||||||||||||||||||||||||||
|
Introduction Line 53 Please add a reference concerning the other approaches used to control PPN |
Reference has been added |
||||||||||||||||||||||||||||||||||||||||||||||||||||||||||||||||||||||||
|
Material and Methods 2.4 How many plants did you use per treatment? Please indicate the number |
All the treatments were conducted in triplicates. It has been mentioned in section 2.4. |
||||||||||||||||||||||||||||||||||||||||||||||||||||||||||||||||||||||||
|
Results Lines 237-242 Expression of gene PR1 was referred to the untreated control. I understand that the value of UC is 1, but no reference is inserted in the text or in the legends of the Figures 1 and 2. I recommend adding a small sentence i.e. "compared to untreated control (set as a unit) at all times" in the legends of figures 1 and 2 or a dotted line at value 1 level, in all graphs, which indicates the expression of the untreated control. |
The y-axis in figure 1 and 2 indicates the mean fold change compared to UC. It’s a ratio. If the ratio was equal to 1, it means the expression was not different from that of UC. If the ratio was more or less than 1, it indicates that the expression was different from that of UC. All of this explanation has been included under ‘statistical analysis’ section. |
||||||||||||||||||||||||||||||||||||||||||||||||||||||||||||||||||||||||
|
Lines 248-249 You are comparing expression in roots between 15dpt and 3dpt. According to the graph 1b the fold change is more than 4-fold higher because the value of 3dpt is set at 0.5. Please recalculate.
|
The values have been recalculated and corrected accordingly |
||||||||||||||||||||||||||||||||||||||||||||||||||||||||||||||||||||||||
|
Line 251 Also in this case the value is not correct. Please check and recalculate |
The values have been recalculated and corrected accordingly |
||||||||||||||||||||||||||||||||||||||||||||||||||||||||||||||||||||||||
|
Figure 1 Please uniform all graphs, some have grid others no.
|
All the grids have been removed to make them look uniform. |
||||||||||||||||||||||||||||||||||||||||||||||||||||||||||||||||||||||||
|
Line 282 You can state significantly lower if there are asterisks.
|
(*) Symbol indicates mean relative fold change which is significantly different from that of the UC. This sentence has been indicated in the legends of figure 1, 2 and 3. |
||||||||||||||||||||||||||||||||||||||||||||||||||||||||||||||||||||||||
|
Line 311 It seems to me that there are mistakes in the calculation of folds of expression in treated tissues respect to IU. Please check and recalculate.
|
The fold of expression have been re-calculated and corrected accordingly within the text |
||||||||||||||||||||||||||||||||||||||||||||||||||||||||||||||||||||||||
|
Figure 5 shows that protein spots 1, 2 and 3 were present in samples extracted from Nemguard, ACS3048 treated plants and untreated control. For corroborating your statement “However, these protein spots were faintly observed even in the untreated control plants” I suggest to add a graph showing the quantitation of the expressed bands
|
Figure 5 shows that protein spots 1,2and 3 were present in all the as indicated with the yellow circles.
We have not performed any gel analysis or densitometry, therefore, quantitation of the expressed bands is not possible at this stage. This will be kept in mind for all our future studies. I have edited the sentence in the manuscript as “However, these protein spots were also observed even in the untreated control plants”.
|
||||||||||||||||||||||||||||||||||||||||||||||||||||||||||||||||||||||||
|
Minor points: Introduction Line 38 Because you already cited the genus Meloidogyne (i.e. Meloidogyne arenaria) you can shorten M. incognita, M. javanica and so on.
|
edited |
||||||||||||||||||||||||||||||||||||||||||||||||||||||||||||||||||||||||
|
Material and Methods 2.1 Line 94 Solanum lycopersicum shoud be in Italic |
edited |
||||||||||||||||||||||||||||||||||||||||||||||||||||||||||||||||||||||||
|
Line 94 Delete “variety”
|
deleted |
||||||||||||||||||||||||||||||||||||||||||||||||||||||||||||||||||||||||
|
Line 95 Better “roots that were infected with M. javanica”
|
changed |
||||||||||||||||||||||||||||||||||||||||||||||||||||||||||||||||||||||||
|
2.2 Line 109 Delete “variety”
|
deleted |
||||||||||||||||||||||||||||||||||||||||||||||||||||||||||||||||||||||||
|
Lines 117-119 I suggest to rewrite the sentence as “…..4-week old tomato plants were treated individually with 3% (v/v) ACS5075 or 3g ACS3048 only once and were harvested after 3, 5, 9 and 15 days”
|
rewritten |
||||||||||||||||||||||||||||||||||||||||||||||||||||||||||||||||||||||||
|
Lines 119-123 I suggest to move this part “Six replications….including the control” immediately after line 126 “…RKN study[17]”
|
edited accordingly |
||||||||||||||||||||||||||||||||||||||||||||||||||||||||||||||||||||||||
|
Line 126 Add a space between RKN and study
|
added |
||||||||||||||||||||||||||||||||||||||||||||||||||||||||||||||||||||||||
|
Line 127 I suggest to delete “After the completion of each treatment…from the soil” because it is a repetition
|
deleted |
||||||||||||||||||||||||||||||||||||||||||||||||||||||||||||||||||||||||
|
Lines 131-132 Delete “All the treatments were conducted with six replications, therefore”
|
deleted |
||||||||||||||||||||||||||||||||||||||||||||||||||||||||||||||||||||||||
|
Line 132 Twelve (now capital letter)
|
changed |
||||||||||||||||||||||||||||||||||||||||||||||||||||||||||||||||||||||||
|
2.5 Line 215 S. lycopersicum
|
changed |
||||||||||||||||||||||||||||||||||||||||||||||||||||||||||||||||||||||||
|
Results Line 250 Better to write “trend” instead of “pattern”
|
Replaced ‘pattern’ with ‘trend’ |
||||||||||||||||||||||||||||||||||||||||||||||||||||||||||||||||||||||||
|
Line 286 This sentence is incomplete
|
edited |
||||||||||||||||||||||||||||||||||||||||||||||||||||||||||||||||||||||||
|
Figure 3 “housekeeping” not “house keeping”
|
changed |
||||||||||||||||||||||||||||||||||||||||||||||||||||||||||||||||||||||||
|
Discussion Lines 446-448 Delete this sentence. It has already written. |
deleted |
||||||||||||||||||||||||||||||||||||||||||||||||||||||||||||||||||||||||
|
Line 456 soybean
|
changed |
||||||||||||||||||||||||||||||||||||||||||||||||||||||||||||||||||||||||
|
Line 473 P. syringae
|
corrected |
||||||||||||||||||||||||||||||||||||||||||||||||||||||||||||||||||||||||
|
Lines 480 and 491 Delete Table 2
|
Deleted |
||||||||||||||||||||||||||||||||||||||||||||||||||||||||||||||||||||||||
|
|
|
||||||||||||||||||||||||||||||||||||||||||||||||||||||||||||||||||||||||

Reviewer 3 Report (New Reviewer)
This manuscript entitled “Investigating the Effects of Alltech Crop Science (ACS) Products on Plant Defence Mechanisms following Root-Knot Nematode Infestation” aimed to evaluate the effect of two Alltech Crop Science products, ACS5075 and ACS3048, on the immune response of tomato plants against root-knot nematodes. The results suggest that ACS5075 enhances the expression of ACO and PR-1 genes, making the plants less sensitive to Meloidogyne javanica infection. To evaluate the expression of these genes, real-time PCR analyses were performed. The results indicated that treatment with ACS5075 significantly increased the expression of ACO and PR-1 genes, both post-treatment and post-infection with M. javanica. The study was conducted with six replicates, and 12 tissue samples were collected per treatment. The results suggest that ACS5075 and ACS3048 can be used to improve the immune response of tomato plants against root-knot nematodes.
Minor suggestions:
The authors need to revise the title of the paper in a more meaningful way. the title is long and has some unnecessary information, suggestion: “Investigating the Effects of Alltech Crop Science Products on Plant Defense Against Root-Knot Nematodes.”
The abstract is written in a way lacks logic. It should highlight the salient findings more critically;
In the materials and methods of the manuscript, in the section where the RT-PCR procedures are described and in the statistical analyses, the authors report that they performed a relative expression using the Livak 2001 method; However, it was not clear in the text who was the calibrating sample used in the relationship parameters. TO DESCRIBE!!!
Authors should discuss the results integrally. The discussion is based on individual results. I suggest that integrating the results will give more value to the work. I suggest that you discuss by integrating all your results. You can use correlation tests (PCA or Pearson Correlation).
The conclusion is totally confusing. Re-write the conclusion! It needs to be much improved.
Author Response
Response to reviewer comments
|
Reviewer 1 |
|
||||||||||||||||||||||||||||||||||||||||||||||||||||||||||||||||||||||||
|
The manuscript “Investigating Effects of Alltech Crop Science (ACS) Products on Plant Defence Mechanisms following Root‐Knot Nematode Infestation” has been improved with the revisions of the reviewers. Some minor revisions are proposed to improve the quality of the manuscript below. |
|
||||||||||||||||||||||||||||||||||||||||||||||||||||||||||||||||||||||||
|
Line 34. Elsen {4} . Those brackets are in journal format? |
Corrected to square brackets |
||||||||||||||||||||||||||||||||||||||||||||||||||||||||||||||||||||||||
|
Line 38. Would you change Meloidogyne to M. ? |
Changed |
||||||||||||||||||||||||||||||||||||||||||||||||||||||||||||||||||||||||
|
Lines 51 to 54. Need a reference |
Added |
||||||||||||||||||||||||||||||||||||||||||||||||||||||||||||||||||||||||
|
Line 66. I would write “among others” because not only Rhizobacterias are responsible for that activation |
Edited accordingly |
||||||||||||||||||||||||||||||||||||||||||||||||||||||||||||||||||||||||
|
Lines 94. Solanum lycopersicum should be in italics |
Changed |
||||||||||||||||||||||||||||||||||||||||||||||||||||||||||||||||||||||||
|
Line 94 and 109 Delete variety |
Deleted |
||||||||||||||||||||||||||||||||||||||||||||||||||||||||||||||||||||||||
|
Line 121. Change days post-treatment by dpt |
Changed |
||||||||||||||||||||||||||||||||||||||||||||||||||||||||||||||||||||||||
|
Line 96. What is a SETU? |
Its South East Technological University (SETU) |
||||||||||||||||||||||||||||||||||||||||||||||||||||||||||||||||||||||||
|
Line 118, I understand, in the solid treatment, you remove the plant, mix it with the product and then put the plant back into the pot?
Detailed product treatments should be explained |
Detailed explanation about product treatments have been written within this paragraph. |
||||||||||||||||||||||||||||||||||||||||||||||||||||||||||||||||||||||||
|
Lone 120-122. What changes between 3dpt and 5dpt and all the other dpt? If it is the same, rewrite the sentence. Proposal “Plants were uprooted after each dpi to proceed with RNA extraction and then RT‐PCR” and then lines 120 – 126 I would remove it. |
Edited accordingly |
||||||||||||||||||||||||||||||||||||||||||||||||||||||||||||||||||||||||
|
Line 129. I would remove (100mg per sample) and explain better the complete sentence. Proposal: “Leaf and root tissue, separately, was ground to a fine powder using liquid nitrogen with a sterile porcelain pestle and mortar, and 100mg per sample was used for RNA extraction immediately or kept at -80ºC until it's used.” |
Edited |
||||||||||||||||||||||||||||||||||||||||||||||||||||||||||||||||||||||||
|
Line 434 and Table 1. Molinari 21, studied JERF3. Is it the same as you study? If it is the same I would change all JERF to JERF3. |
Have been changed |
||||||||||||||||||||||||||||||||||||||||||||||||||||||||||||||||||||||||
|
Line 456. Change soyabean cyst nematodes to soybean cyst nematodes |
changed |
||||||||||||||||||||||||||||||||||||||||||||||||||||||||||||||||||||||||
|
Ine477. Change possibly to possible |
changed |
||||||||||||||||||||||||||||||||||||||||||||||||||||||||||||||||||||||||
|
Table 1. Line 219. I propose to include a column with the references and not in the footnote |
A column with references have been included and footnote deleted |
||||||||||||||||||||||||||||||||||||||||||||||||||||||||||||||||||||||||
|
Figure 1. needs to be improved. Figure 2. needs to be improved.
Figure 3 needs to be improved. Foot notes look blurry
|
Both Figure 1 and 2 have been improved and have been uploaded as a separate word file named “Figure 1, 2 and 3”
Figure 3 has been improved and the clarity of foot notes has been improved.
All Figure 1, 2 and 3 have been improved and have been uploaded as a separate word file named “Figure 1, 2 and 3” |
||||||||||||||||||||||||||||||||||||||||||||||||||||||||||||||||||||||||
|
Reviewer 3 |
|
||||||||||||||||||||||||||||||||||||||||||||||||||||||||||||||||||||||||
|
The authors need to revise the title of the paper in a more meaningful way. the title is long and has some unnecessary information, suggestion: “Investigating the Effects of Alltech Crop Science Products on Plant Defense Against Root-Knot Nematodes.”
|
Title has been changed as per the suggestion |
||||||||||||||||||||||||||||||||||||||||||||||||||||||||||||||||||||||||
|
The abstract is written in a way lacks logic. It should highlight the salient findings more critically; |
Abstract has been edited |
||||||||||||||||||||||||||||||||||||||||||||||||||||||||||||||||||||||||
|
|
|
||||||||||||||||||||||||||||||||||||||||||||||||||||||||||||||||||||||||
|
In the materials and methods of the manuscript, in the section where the RT-PCR procedures are described and in the statistical analyses, the authors report that they performed a relative expression using the Livak 2001 method; However, it was not clear in the text who was the calibrating sample used in the relationship parameters. TO DESCRIBE!!!
|
The cycle threshold (CT values) that were generated by the Roche machine were recorded individually. ΔCT values were calculated by deducting the individual CT values from that of the CT values obtained for the respective Actin gene per each sample [ΔCT(a target sample)−ΔCT(Actin)]. Using a reference gene (Actin) as a standard, the ultimate outcome of this method is represented as the mean relative fold change in target gene expression in a target sample compared to untreated control, which acted as a calibrating sample.
The above description has been added in section 2.3 under materials and methods. |
||||||||||||||||||||||||||||||||||||||||||||||||||||||||||||||||||||||||
|
Authors should discuss the results integrally. The discussion is based on individual results. I suggest that integrating the results will give more value to the work. I suggest that you discuss by integrating all your results. You can use correlation tests (PCA or Pearson Correlation).
|
Post-inoculation with M. javanica and treatment with ACS5075, in the root tissues, a positive correlation (r=0.4 and 0.2) has been noted while correlating expression of PR-1 and ACO genes, PR-5 and JERF 3 genes, respectively. This means with an increase in expression of one gene there is a positive increase of fixed proportion in the expression of other genes.
In the leaf tissues, a strong positive correlation (r=0.8, 0.7, 0.6) was also noted while correlating ACO and JERF 3, JERF 3 and PR-1, ACO and PR-1 genes, respectively.
All of the above has been added under the discussion section. |
||||||||||||||||||||||||||||||||||||||||||||||||||||||||||||||||||||||||
|
The conclusion is totally confusing. Re-write the conclusion! It needs to be much improved. |
Conclusion part has been re-written |
||||||||||||||||||||||||||||||||||||||||||||||||||||||||||||||||||||||||
|
Reviewer 2 |
|
||||||||||||||||||||||||||||||||||||||||||||||||||||||||||||||||||||||||
|
Introduction Line 53 Please add a reference concerning the other approaches used to control PPN |
Reference has been added |
||||||||||||||||||||||||||||||||||||||||||||||||||||||||||||||||||||||||
|
Material and Methods 2.4 How many plants did you use per treatment? Please indicate the number |
All the treatments were conducted in triplicates. It has been mentioned in section 2.4. |
||||||||||||||||||||||||||||||||||||||||||||||||||||||||||||||||||||||||
|
Results Lines 237-242 Expression of gene PR1 was referred to the untreated control. I understand that the value of UC is 1, but no reference is inserted in the text or in the legends of the Figures 1 and 2. I recommend adding a small sentence i.e. "compared to untreated control (set as a unit) at all times" in the legends of figures 1 and 2 or a dotted line at value 1 level, in all graphs, which indicates the expression of the untreated control. |
The y-axis in figure 1 and 2 indicates the mean fold change compared to UC. It’s a ratio. If the ratio was equal to 1, it means the expression was not different from that of UC. If the ratio was more or less than 1, it indicates that the expression was different from that of UC. All of this explanation has been included under ‘statistical analysis’ section. |
||||||||||||||||||||||||||||||||||||||||||||||||||||||||||||||||||||||||
|
Lines 248-249 You are comparing expression in roots between 15dpt and 3dpt. According to the graph 1b the fold change is more than 4-fold higher because the value of 3dpt is set at 0.5. Please recalculate.
|
The values have been recalculated and corrected accordingly |
||||||||||||||||||||||||||||||||||||||||||||||||||||||||||||||||||||||||
|
Line 251 Also in this case the value is not correct. Please check and recalculate |
The values have been recalculated and corrected accordingly |
||||||||||||||||||||||||||||||||||||||||||||||||||||||||||||||||||||||||
|
Figure 1 Please uniform all graphs, some have grid others no.
|
All the grids have been removed to make them look uniform. |
||||||||||||||||||||||||||||||||||||||||||||||||||||||||||||||||||||||||
|
Line 282 You can state significantly lower if there are asterisks.
|
(*) Symbol indicates mean relative fold change which is significantly different from that of the UC. This sentence has been indicated in the legends of figure 1, 2 and 3. |
||||||||||||||||||||||||||||||||||||||||||||||||||||||||||||||||||||||||
|
Line 311 It seems to me that there are mistakes in the calculation of folds of expression in treated tissues respect to IU. Please check and recalculate.
|
The fold of expression have been re-calculated and corrected accordingly within the text |
||||||||||||||||||||||||||||||||||||||||||||||||||||||||||||||||||||||||
|
Figure 5 shows that protein spots 1, 2 and 3 were present in samples extracted from Nemguard, ACS3048 treated plants and untreated control. For corroborating your statement “However, these protein spots were faintly observed even in the untreated control plants” I suggest to add a graph showing the quantitation of the expressed bands
|
Figure 5 shows that protein spots 1,2and 3 were present in all the as indicated with the yellow circles.
We have not performed any gel analysis or densitometry, therefore, quantitation of the expressed bands is not possible at this stage. This will be kept in mind for all our future studies. I have edited the sentence in the manuscript as “However, these protein spots were also observed even in the untreated control plants”.
|
||||||||||||||||||||||||||||||||||||||||||||||||||||||||||||||||||||||||
|
Minor points: Introduction Line 38 Because you already cited the genus Meloidogyne (i.e. Meloidogyne arenaria) you can shorten M. incognita, M. javanica and so on.
|
edited |
||||||||||||||||||||||||||||||||||||||||||||||||||||||||||||||||||||||||
|
Material and Methods 2.1 Line 94 Solanum lycopersicum shoud be in Italic |
edited |
||||||||||||||||||||||||||||||||||||||||||||||||||||||||||||||||||||||||
|
Line 94 Delete “variety”
|
deleted |
||||||||||||||||||||||||||||||||||||||||||||||||||||||||||||||||||||||||
|
Line 95 Better “roots that were infected with M. javanica”
|
changed |
||||||||||||||||||||||||||||||||||||||||||||||||||||||||||||||||||||||||
|
2.2 Line 109 Delete “variety”
|
deleted |
||||||||||||||||||||||||||||||||||||||||||||||||||||||||||||||||||||||||
|
Lines 117-119 I suggest to rewrite the sentence as “…..4-week old tomato plants were treated individually with 3% (v/v) ACS5075 or 3g ACS3048 only once and were harvested after 3, 5, 9 and 15 days”
|
rewritten |
||||||||||||||||||||||||||||||||||||||||||||||||||||||||||||||||||||||||
|
Lines 119-123 I suggest to move this part “Six replications….including the control” immediately after line 126 “…RKN study[17]”
|
edited accordingly |
||||||||||||||||||||||||||||||||||||||||||||||||||||||||||||||||||||||||
|
Line 126 Add a space between RKN and study
|
added |
||||||||||||||||||||||||||||||||||||||||||||||||||||||||||||||||||||||||
|
Line 127 I suggest to delete “After the completion of each treatment…from the soil” because it is a repetition
|
deleted |
||||||||||||||||||||||||||||||||||||||||||||||||||||||||||||||||||||||||
|
Lines 131-132 Delete “All the treatments were conducted with six replications, therefore”
|
deleted |
||||||||||||||||||||||||||||||||||||||||||||||||||||||||||||||||||||||||
|
Line 132 Twelve (now capital letter)
|
changed |
||||||||||||||||||||||||||||||||||||||||||||||||||||||||||||||||||||||||
|
2.5 Line 215 S. lycopersicum
|
changed |
||||||||||||||||||||||||||||||||||||||||||||||||||||||||||||||||||||||||
|
Results Line 250 Better to write “trend” instead of “pattern”
|
Replaced ‘pattern’ with ‘trend’ |
||||||||||||||||||||||||||||||||||||||||||||||||||||||||||||||||||||||||
|
Line 286 This sentence is incomplete
|
edited |
||||||||||||||||||||||||||||||||||||||||||||||||||||||||||||||||||||||||
|
Figure 3 “housekeeping” not “house keeping”
|
changed |
||||||||||||||||||||||||||||||||||||||||||||||||||||||||||||||||||||||||
|
Discussion Lines 446-448 Delete this sentence. It has already written. |
deleted |
||||||||||||||||||||||||||||||||||||||||||||||||||||||||||||||||||||||||
|
Line 456 soybean
|
changed |
||||||||||||||||||||||||||||||||||||||||||||||||||||||||||||||||||||||||
|
Line 473 P. syringae
|
corrected |
||||||||||||||||||||||||||||||||||||||||||||||||||||||||||||||||||||||||
|
Lines 480 and 491 Delete Table 2
|
Deleted |
||||||||||||||||||||||||||||||||||||||||||||||||||||||||||||||||||||||||
|
|
|
||||||||||||||||||||||||||||||||||||||||||||||||||||||||||||||||||||||||

This manuscript is a resubmission of an earlier submission. The following is a list of the peer review reports and author responses from that submission.
Round 1
Reviewer 1 Report
The manuscript entitled “Investigating Effects of Alltech Crop Science (ACS) Products on Plant Defence Mechanisms following Root‐Knot Nematode Infestation” falls under the scope of the Microorganisms journal and it addresses an essential subject in the new paradigm of reduction of chemical nematicides looking for new alternatives and the most important how those new products work.
However, it needs major revisions to be published. The writing of the manuscript is very poor and it has not the quality for publishing if changes are not done.
Underneath the items that should be considered to improve the quality of the manuscript:
General items
References must be numbered in order of appearance in the texts
Line 5. In my opinion, the research group should be in line 7
Summary
Line 21 Write PR‐1 and ACO in the same order in all the summary
Line 21. Results of the reproduction of the nematode are not presented, so remove it from the summary.
Title
I recommend to change by “Investigating the effects” or “Plant Defence Mechanisms Inducted by Alltech Crop Science (ACS) following Root‐Knot Nematode Infestation”
Introduction
Line 36. It should be eggplants instead of eggplant
Line 37. Cysts nematode don't do malformations. Please change the sentence
Line 39. Widely spread is repeated in some lines above. Please change it
Line 40. (Zakaria et al., 2013 reference did not do any investigation about the widely spread of Meloidogyne, only testing control methods. Rewrite the sentence or change the reference.
Line 45. Needs a reference
Line 46. Douda et al., 2021 did not investigate the effects on human health, ground water, and animals. Rewrite the sentence or change the reference.
Line from 47 to 55. The authors only talk about biofumigation as an alternative control method against PPN. Is there a reason? Other methods are not used or developed in the last 25 years? Also, biofumigation is an effective method against PPN and the authors show only the negative. In addition, biofumigation is not only carried out by Brassicaceae family. Please reconsider rewrite the sentence.
Line 57. A better reference can be found instead of Twamley et al., 2019. For example https://www.frontiersin.org/articles/10.3389/fpls.2014.00488/full or https://www.annualreviews.org/doi/abs/10.1146/annurev-arplant-042916-041132
Line 57. But priming gives the plant rubosity and makes the plant respond better to an abiotic or biotic factor. Then High Crop productivity is a consequence of priming
Line 63 and 64 Sentence and definition are not correct
Line 66. What about microorganisms or insects? https://www.frontiersin.org/articles/10.3389/fpls.2020.00796/full
From lines 66 to 74, better provide examples of PPN or RKN which are the targets of the manuscript.
Line 85 AND have not to be in italics
You should reconsider rewrite the introduction
Material and methods
Line 90. Change Moneymaker variety by cv. Moneymaker
Line 92. What freshly infected means?
Line 92. Were the plants in individual pots?
Line 92-93. Do tomato roots arrive at 4-5 cm after 3-4 weeks of sowing?
Line 95. Neudorf is a brand; which pesticide did you use?
Line 95. What does Once in 3 weeks means?. Every 3 weeks or 3 weeks after sowing?
Line 99. Application is not the correct word,
Line 99. I would change PPN by RKN
Line 100. Change was by were.
Line 104. Solanum lycopersicum should be in line 90 and in italics
Line 104. Which cultivar did you use?
Line 105. Delete relative humidity
Line 106. What is garden soil? Was it sterilized?
Line 107. A space is needed between conductivity and the value.
Line 110. I propose to write the capacity of the pots.
Line 111. Which soil did you use? Was it sterilized?
Line 116. Use RKN
Line 119. After all the treatments?
2.2. FIRST SET OF EXPERIMENTS ARE NOT CLEAR
Line113. Treatments were done 3,5 9 and 15 days from transplanting?
How the products were applied? The 3% is v/v as it is explained in line 170? How the solid one was applied? Diluted or mixed into the soil?
Leaves were removed after 3 days and the plants were left without leave until day nine. That needs to be clear because is not well explained
Line 125 to 128. Consider rewriting to better understanding
Line 140. After 15 dpi, what does it mean? How many days?
Line 161. That you do RT-PCR of all the plant tissues doesn’t need to be repeated.
Line 169. Add that Nemguard is a commercial nematicide based on garlic.
Line 170. Di you treat for 9 days, every day? Or only the day 9dpt?
Line 172. Write -80
Line174. Remove “per”
Line 175. What triplicate means?
Line 176. Is “on to” correct?
Line 203. What is PVDF?
Line 204. What is PTC?
Line 214. Lycopersicum esculentum is the old Latin name for tomato. Please change it
Line 222. Why do you use standard variation instead of error standard?
In the second set of experiments, in line 228, the data was also as the first one, mean and standard variation. So please write it.
Line 231 “and” should not be in italics
From 233 until the end should be in the results part
Results
Line 244,258. UC is not explained in the Material and methods section
Line 245. Needs to be added ACS before 3048
Line 270. Are you sure is Figure 3d?
Line 292. It is not clear the difference between the Fig 3a and Fig 3b.
Line 293, 311 “and” should not be in italics
Line 299, 302, 304, 308, 3010 IC should be IU?
Line 300. Change was per were
Line 316. Change was per is
Line from 337 until 341 is already explained in Material and methods
Line 343. Change Figure 3 per Figure 4
Line 350 and 358. add TM after nemguard
Why there are some circles in Figure 4?
Line 355. Change have been per were
Table 2. Rewrite the title to explain more the table
Discussion
Line 379 change PPN to RKN
Text from lines 380 to 382 is repeated in the introduction. Needs to be removed
From line 384 to 389 need references
Line 407, reference is not well placed because Derksen doesn’t study RKN
Line 422. Mention what is P of P. syringae
Line 439 Change Meloidogyne javanica by M. javanica
Line 457 needs a reference
Line 463. The reduction of M. javanica needs a reference
Acknowledgments
Line 483. “their” should be “this”?
Line485. “Meloidogyne” should be in italics
Dr and Dr. are written. Be consistent
References
References should be in journal format (https://www.mdpi.com/journal/microorganisms/instructions#references)
Names of plants and microorganisms should be in italics
Figures
Images should be in higher quality. They look blurry.
In Figure 1 doesn’t appear the lines of the graph. Please include them
In Figure 2 doesn’t appear the lines of the graph. Please include them
The names of the products are not correct in Line 325. Table 6 doesn’t exist
Be consistent in the figure and the text because they are different.
Figure 3 needs to be more work. Some bars are behind names and it doesn’t have the same size of letter. Some of the text looks blurry.
It is not clear the difference between the Fig 3a and Fig 3b. Make it clear
Reviewer 2 Report
The manuscript provides some information on the function of two ACS products that present potential use for plant parasitic nematode control. Overall, the manuscript is well thought out, easy to read and understand. Results and methodology are appropriate. I have just a concern about the protein spots described and the functions associated.
Some minor comments need consideration.
Introduction
Page 2 line 79: I suggest to eliminate Pulavarty et al. 2022 because you are describing the effects of ACS products on tomato roots/plants and the manuscript by Pulavarty et al. 2022 is based on results obtained from treatments in potato.
Material and Methods
Page 3 line 138 Do you mean “development of galls on the roots”? Fifteen days seems to me a short time to find egg masses.
Results:
Figure 4: What do the red circles indicate?
Discussion
Figure 5 shows that protein spots 1, 2 and 3 were present in samples extracted from Nemguard, ACS3048 treated plants and untreated control.
I think that if these proteins are present in the untreated control they cannot be associated to the treatments. Therefore, the argumentation that the presence, for example, of disease resistance protein RPP13-like could be one of the main reasons for reduced Meloidogyne javanica infestation in ACS treated tomato plants is not correct. Likewise, proteins from spot 2 and 3 are also present in untreated controls suggesting their involvement in the physiological processes of plants. Their function in promoting increased leaf biomass or root length upon treatment is a simple speculation. You are showing what happen in the leaves, but very interesting would be the analysis of proteins in roots, the targets of nematodes.
I suggest to rewrite this part of the discussion.
I find interesting that plants treated with ACS5075 lack protein spots 3 and 4. Can you associate this deficiency with their phenotype? Please try to explain.
Typoes:
Page 2 line 60 “through”
Page 4 line 193 “onto”
Page 7 Please check the acronym IC along the paragraph gene expression analysis. I suppose you mean IU (inoculated untreated plants)
Page 10 line 343 Figure 4
Page 11 line 345 Figure 4
Page 13 line 423 “the activity of…”